# Prevalence of chronic kidney disease among young people living with HIV in Sub Saharan Africa: A systematic review and meta-analysis

Esther M. Nasuuna[1,2]*, Nicholus Nanyeenya[3], Davis Kibirige[1,4], Jonathan Izudi[2,5], Chido Dziva Chikwari[6,7], Robert Kalyesubula[1,8], Barbara Castelnuovo[2], Laurie A. Tomlinson[9], Helen A. Weiss[7]

1 Medical Research Council/Uganda Virus Research Institute and London School of Hygiene and Tropical Medicine Uganda Research Unit, Noncommunicable Diseases Program, Entebbe, Uganda, 2 Infectious Diseases Institute, Makerere University, College of Health Sciences, Kampala, Uganda, 3 Department of Epidemiology and Biostatistics, School of Public Health, Makerere University College of Health Sciences, Kampala, Uganda, 4 Department of Medicine, Uganda Martyrs Hospital Lubaga, Kampala, Uganda, 5 Department of Community Health, Mbarara University of Science and Technology, Mbarara, Uganda, 6 Biomedical Research and Training Institute, Harare, Zimbabwe, 7 MRC International Statistics and Epidemiology Group, London School of Hygiene & Tropical Medicine, London, United Kingdom, 8 Departments of Physiology and Medicine, Makerere University College of Health Sciences, Kampala, Uganda, 9 Department of non-Communicable Disease Epidemiology, London School of Hygiene and Tropical Medicine, London, United Kingdom

* enasuuna@idi.co.ug

**Data Availability Statement:** All relevant data are within the manuscript and its Supporting Information files.

## Abstract

### Background

Globally, the prevalence of chronic kidney disease (CKD) is increasing among young people living with HIV (YPLHIV), with inconsistent estimates. Aggregated data on the prevalence of CKD are needed in sub-Saharan Africa (SSA) to inform strategies for early diagnosis and management. We conducted a systematic review and meta-analysis to estimate the pooled prevalence of CKD among YPLHIV in SSA.

### Methods

We searched Medline/PubMed, EMBASE, African Index Medicus, and African Journals Online for articles reporting the prevalence of CKD among YPLHIV in SSA using predefined search strategies up to 15th January 2024. The reference lists of identified articles were checked for additional eligible studies. The eligibility criteria were studies among YPLHIV aged 10–24 years reporting CKD prevalence defined by either glomerular filtration rate (GFR), albumin-to-creatinine ratio (ACR) or proteinuria. We used a narrative synthesis to report differences between the included studies. The DerSimonian-Laird random effects model was used to pool the CKD prevalence, and heterogeneity was assessed using the Cochrane Q-test and I-squared values. We assessed the risk of bias in each article using the Joanna Briggs Institute checklist and publication bias in a funnel plot and Egger's test.

**Funding:** The author(s) received no specific funding for this work.

**Competing interests:** The authors have declared that no competing interests exist.

## Results

Of 802 retrieved articles, 15 fulfilled the eligibility criteria and were included in the meta-analysis. Of these, 12 (80%) were cross-sectional studies that used estimated GFR to diagnose CKD. Only one study followed the standard definition of CKD. The pooled CKD prevalence from 15 studies was 12% (95% CI 6.0–19.5%), ranging from 0.8% to 53.1% according to the definition used, with a high degree of heterogeneity ($I^2$ = 97.7%, p<0.001). The included studies were of moderate quality, with no evidence of publication bias. Sensitivity analysis showed that the findings were robust to the methodological and analytic approach.

## Conclusion

CKD prevalence among YPLHIV is moderately high and highly heterogeneous across SSA. The standard definition of CKD should be used to enable estimation of CKD prevalence in different studies and settings. HIV programs enrolling YPLHIV should routinely screen for CKD to ensure early diagnosis and management.

## Trial registration

**PROSPERO registration number**: CRD42022347588.

## Introduction

In 2022, an estimated 3.4 million young people (15–24 years) and 1.7 million adolescents (10–19 years) were living with human immunodeficiency virus (HIV) globally, of whom approximately 80% lived in sub-Saharan Africa (SSA) [1–3]. The number of young people living with HIV (YPLHIV) in SSA is increasing due to increased infant survival [4–6]. Among children living with HIV, there has been a 10-fold reduction in mortality and improved survival since 2004 largely due to the scale-up of antiretroviral therapy (ART) [4]. As a result, infants who acquire HIV perinatally survive into adulthood and develop comorbidities such as chronic kidney disease (CKD), cardiovascular diseases, and type 2 diabetes [7,8]. These comorbidities affect almost all body systems and have important implications for HIV treatment, quality of life, and survival [6,9].

CKD is becoming more prevalent globally and is projected to be the fifth leading cause of mortality globally by 2040 [10,11]. In both adults and children, CKD is defined by Kidney Disease Improving Global Outcomes (KDIGO) as a glomerular filtration rate (GFR) <60ml/min/1.73m$^2$ and/or the presence of markers of kidney damage for three or more months [12,13]. GFR can be estimated (eGFR) using a biomarker and one of the estimating equations or measured (mGFR) using compounds such as Iohexol [14–16]. Markers of kidney damage include structural abnormalities seen on imaging (changes in kidney size and increased echogenicity), histology, electrolyte disorders, urine sediment abnormalities, proteinuria, and albuminuria [12]. The Global Burden of Disease study (2017) estimates the prevalence of CKD at 9.1% (95% confidence interval (CI) 8.5–9.8%) globally [17] varying by region: 6.0–17.0% in the United States, 7.0–34.3% in Asia, 3.3–17.3% in Europe [17] and 13.9% (95% CI 12.2–15.7) in SSA [18]. The rising prevalence of CKD in high-income countries is partially attributed to the increasing incidence of diabetes mellitus and hypertension [19], while in several low- and middle-income countries (LMICs), HIV and Hepatitis C are the major contributing factors [18].

Among people living with HIV, the kidney is affected by direct HIV infection of the renal epithelial cells, deposition of immune complexes and toxicities from drugs used to treat HIV or opportunistic infections. These include Tenofovir Disoproxil Fumarate (TDF), protease inhibitors and amphotericin B [20–22]. Although ART initiation is protective against kidney diseases, especially HIV-associated nephropathy [23], there is evidence of increased prevalence of kidney diseases after ART initiation [24–26]. CKD is among the 10 most common non-infectious complications of HIV in the United States of America [20,27].

YPLHIV are at an increased risk for CKD compared to those without HIV [28]. Young people who acquire HIV through vertical transmission have a fourfold higher risk of CKD compared to other young people without HIV or those infected later in life, due to chronic HIV infection of an immature kidney, long-term ART exposure, and drug toxicities [29]. CKD is particularly difficult to estimate in young people due to dependence on eGFR using serum creatinine which is influenced by muscle mass that changes with age and nutritional status [30,31]. Creatinine levels in children are also harder to measure as they are lower than those in adults and require a highly sensitive test [32]. Other reasons for underestimation of CKD prevalence are a lack of surveillance and poor measurement and reporting of CKD [29,33]. CKD prevalence reported from several observational studies in SSA is inconsistent.

Our aim was to conduct the first systematic review and meta-analysis to determine the pooled prevalence of CKD among YPLHIV in SSA and to understand the reasons for possible heterogeneity in the prevalence.

## Methods

The Preferred Reporting Items for Systematic Reviews and Meta-Analyses (PRISMA) statement 2020 was used to guide the systematic review (S1 Appendix) [34]. The study protocol (S2 Appendix) was registered with Prospective Register of Systematic Reviews (PROSPERO) and was assigned the registration number CRD42022347588.

### Eligibility criteria

Inclusion criteria were manuscripts published in any year reporting the prevalence of CKD using any predefined definition that shows kidney damage or kidney injury, including the Kidney Disease Improving Global Outcomes (KDIGO) definitions (Box 1), among YPLHIV in any of the sub-Saharan African countries, whose population had a mean or median age between 10 and 24 years. We excluded case-control studies, case series, case reports, conference abstracts without accompanying manuscripts, and studies of low quality (those that scored less than 50% on assessment).

---

### Box 1. KDIGO CKD Definition

CKD KDIGO definition: GFR <60 ml/min per $1.73m^2$ or markers of kidney damage for >3 months. Markers of kidney damage: Albumin creatinine ratio (ACR) >30mg/g, Protein creatinine ratio (PCR) >150mg/g [35]

#### Five stages of chronic kidney disease

Stage 1 with normal or high GFR (GFR > 90 mL/min)

Stage 2 Mild CKD (GFR = 60–89 mL/min)

---

Stage 3A Moderate CKD (GFR = 45–59 mL/min)

Stage 3B Moderate CKD (GFR = 30–44 mL/min)

Stage 4 Severe CKD (GFR = 15–29 mL/min)

Stage 5 End Stage CKD (GFR <15 mL/min)

https://www.davita.com/education/kidney-disease/stages

The outcome of interest was the prevalence of CKD computed as a proportion with a 95% confidence interval (CI). The diagnosis of CKD could be based on either estimated or measured GFR, or an albumin creatinine ratio (ACR) $\geq$30mg/g or proteinuria.

## Information sources and search strategy

The main search was conducted on 30[th] June 2022 with an update on 15[th] January 2024. We searched four databases (Medline via PubMed, EMBASE, African Index Medicus, and African Journals Online) for eligible articles without restrictions on the year of publication or language. Search terms were developed for each database, including Medical Subject Heading (MESH) terms and keywords (S3 Appendix). Search terms included i) chronic kidney diseases, kidney diseases, renal diseases, renal insufficiency, proteinuria, albuminuria); ii) HIV, human immunodeficiency virus; iii) young people, young adults, adolescents, and paediatric; and iv) Africa, SSA, sub-Saharan Africa, Africa south of the Sahara, and individual SSA countries. Reference lists from eligible articles were searched for further relevant articles. A citation search was conducted in the Web of Science to identify other relevant articles.

## Study selection and data collection

Two reviewers (EMN and JI) conducted the database searches and extracted all potentially relevant articles to Endnote 20 (Clarivate, Philadelphia PA). Duplicated articles as determined by title, authors, and journal name were excluded and the remaining articles were uploaded to Rayyan software (https://www.rayyan.ai/) for further de-duplication and screening [36]. Three reviewers (EMN, NN, and DK) independently screened all the articles by title and abstract according to the eligibility criteria. A full-text search by two reviewers (EMN and NN) was conducted to identify articles that fulfilled the eligibility criteria. Two studies published in languages other than English were translated online before review. Discrepancies were resolved by discussion and consensus among all reviewers. If consensus could not be reached, the decision was made by the third senior reviewer (DK). Reference lists and articles citing the included articles were reviewed using Web of Science to identify any relevant articles that had been missed during the database search.

## Data abstraction

EMN performed the initial data abstraction in a Microsoft Excel sheet according to a predetermined data abstraction tool, which was verified by NN for completeness and accuracy. The data abstracted included the first author's name, country of origin, population, inclusion and exclusion criteria, demographic characteristics such as age and sex, sample size, number of participants with CKD, country, region of SSA (Eastern, Western, Central, and Southern), disease definition, equation used, diagnosis, whether or not the diagnosis was confirmed after the required three months, and any comorbidities.

## Study quality assessment

The Joanna Briggs Institute (JBI) checklist for prevalence studies was used to assess the quality of the included studies [37]. A decision to include, exclude, or seek further information was reached for each of the studies. A study was considered low quality if it scored less than 50%, moderate quality if it scored between 50 and 70% and high quality if it scored above 70% [38]. Low quality studies were to be excluded as per the eligibility criteria.

## Data analysis

Statistical analysis was performed in STATA statistical software version 17 (College Station, TX). The Microsoft Excel sheet containing the data items was imported into STATA for analysis. Pooled CKD prevalence was calculated by performing a meta-analysis of proportions using the DerSimonian-Laird random-effects model allowing for generalised linear mixed models with logit link to account for the within study uncertainties [39]. CKD prevalences were displayed in a forest plot. Evidence for the presence of heterogeneity was determined using the Cochrane Q test and quantified using $I^2$ values. If the degree of heterogeneity was above 50%, the studies were considered heterogeneous; otherwise, they were considered non heterogenous [40]. We extended the same model to assess the source of heterogeneity in a sub-group analysis to determine if the pooled prevalence differed by region, definition of CKD used, sample size, whether or not the measurements were repeated or sampling strategy. Furthermore, we performed meta-regression analysis to determine the source of the observed statistical heterogeneity [41]. A funnel plot and Egger's test were used to assess for publication bias [42]. We interpreted a symmetrical funnel plot as suggestive of no publication bias and an asymmetrical one as indicating publication bias. To distinguish between publication bias and small study effect, we performed a contour-enhanced funnel plot. If publication bias was confirmed, we performed a trim and fill analysis. A sensitivity analysis was conducted using the leave-one-out Jacknife method to determine the robustness of the study findings to the analytic approach [43].

The narrative synthesis framework was used to report the differences between the studies. This was based on the Institute of Medicine standards for qualitative synthesis in systematic reviews [44]. Specifically, the methodological characteristics such as sample size, participant inclusion and exclusion criteria, strengths and limitations of each study, potential bias, and patterns across the studies were all reported [44].

# Results

## Study selection

We retrieved a total of 802 articles that reported on chronic kidney disease, chronic renal insufficiency, and any other kind of kidney abnormality among YPLHIV in Africa as per the search strategy and excluded 198 duplicates. These studies were published between 1990 and 2023. After screening the remaining 604 articles (S5 Appendix) by title and abstract, 120 were found to be potentially eligible. Eight of these were duplicates, and 4 had inaccessible full texts, so 108 articles were retrieved for the full-text search. Of these, 93 were excluded for the following reasons: 36 had an ineligible population, 34 did not have the prevalence disaggregated by age for adults and young people, 20 were case–control studies or case reports and 3 were conference abstracts with no accompanying manuscript. Overall, 15 articles fulfilled the inclusion criteria and were included in the meta-analysis (Fig 1).

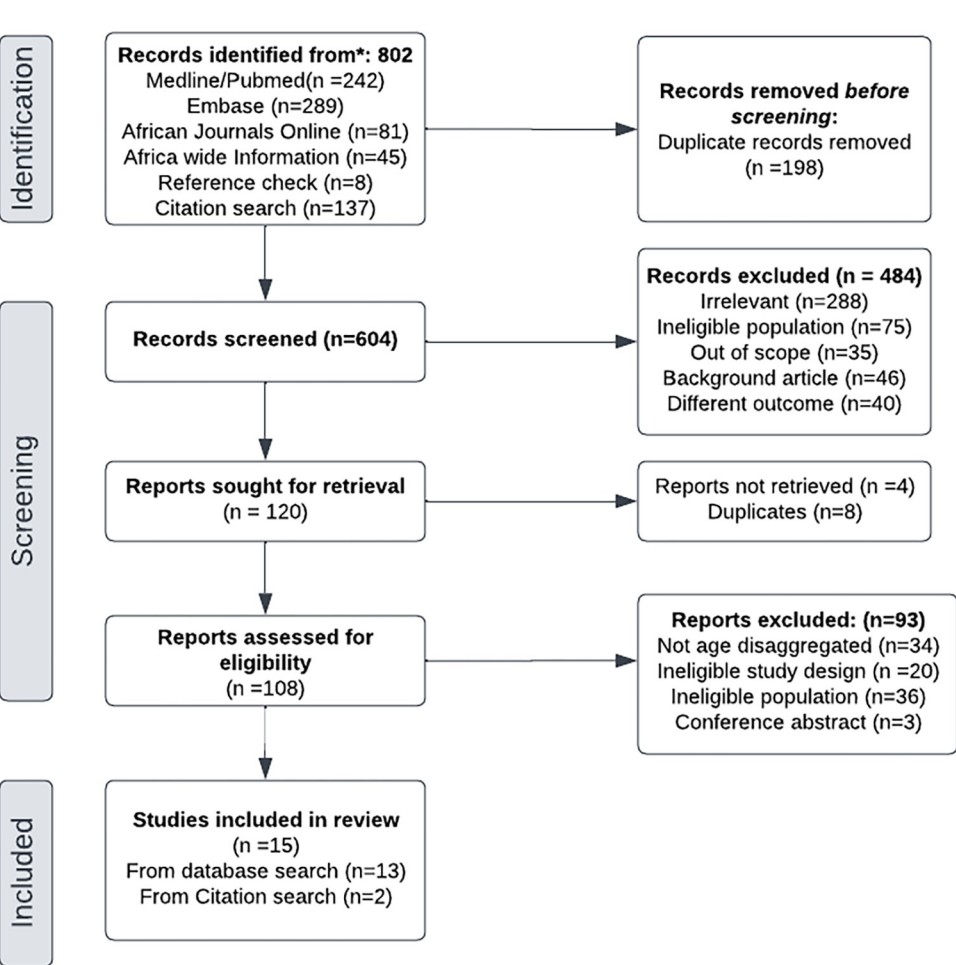

**Fig 1. Flow diagram showing the included studies.**

## Study characteristics

Table 1 summarises the characteristics of the 15 included studies. Details are provided in S4 Appendix. The studies reported information on 4,272 participants from 31 ART clinics in 13 SSA countries. Four studies were from West Africa [45–48], four from East Africa [49–52], six from Southern Africa [6,33,53–56] and one from Central Africa [57]. Eligible studies were conducted between 2013 and 2023. Thirteen studies were cross-sectional [6,33,45–50,52–57], with three nested within cohorts [6,33,51] and two were cohort studies [48,51]. Sample sizes ranged from 12–784 participants (median 266; interquartile range 150–384). Only one study reported sample size estimation and the assumptions made [55].

The mean or median age in each study was between 10 and 24 years, with an age range of 6–30 years. The proportion of female participants in each study was 40–58%. Three studies recruited participants who were ART-naïve [47,48,58], 10 studies recruited ART-experienced participants [6,33,45,46,51–53,55–57], and two studies recruited both ART-naïve and ART-experienced participants [49,50]. Three studies recruited only YPLHIV who were perinatally

**Table 1. Shows details of the included studies.**

| First author, year | Country | Region | Population | Study design | Age (years) | Sex % Female | Sample size | Prevalence (95% CI) | Disease definition /diagnosis |
|---|---|---|---|---|---|---|---|---|---|
| **Diack 2020 [46]** | Burkina Faso, Côte d'Ivoire, Burundi, Senegal, Mali and Cameroon | Central and Western | O to 18 yrs on TDF for 1.5 years | Cross sectional | Median 15.5 (IQR 14–16.8) | 51% | 358 | 4.1% (2–7%) | eGFR 30–60 ml/min/1.73m$^2$ |
| **Okafor 2016 [47]** | South South Nigeria | Western | ART naïve 18 to 81 yrs | Cross sectional | 18–29 yrs | 58% | 96 | 53.1% (43–63%) | eGFR <60 mL/min/1.73m$^2$ |
| **Ekulu 2019 [57]** | Democratic Republic of Congo | Central | <18 yrs on ART | Cross sectional | Mean 11.6 (SD 4.1) | 51% | 401 | 6.5% (4–9%) | eGFR <60ml/min/1.73m$^2$ |
| **Frigati 2019 [6]** | South Africa | Southern | Perinatally infected 9 to 14 yrs on ART > 6 months | Cross sectional | Mean 12 (SD 1.7) | 48% | 384 | 2.3% (1.1–4.4%) | eGFR<90 mL/min/1.73m2 |
| **Frederick 2016 [50]** | Tanzania | Eastern | Both ART naïve and on ART | Cross sectional | 10–14 yrs | 44.6% | 86 | 22.1% (13.9–2.3%) | ACR > = 30 mg/g |
| **Frigati 2018 [33]** | South Africa | Southern | 9 to 14 yrs on ART > 6 months | Cross sectional | 9–14 yrs | 48.7% | 511 | 8.4% (6.1–11.2%) | ACR >30mg/g |
| **Iduoriyekemwen, 2013 [45]** | Nigeria | Western | 10m to 17 yrs on ART | Cross sectional | 12-17yrs | 40% | 12 | 16.7% (2–48%) | Proteinuria 1 + microalbuminuria of ≥20 mg/g eGFR <60ml/min/1.73m$^2$ |
| **Mashingaidze-Mano, 2020 [53]** | Zimbabwe | Southern | <18 yrs on TDF for >6 months | Cross sectional | Median 15 (IQR 13–16) | 44.90% | 198 | 35.9% (29–43%) | eGFR<90ml/min/1.73m$^2$ |
| **Drak, 2021 [54]** | Zimbabwe | Southern | ART naïve 12 to 17 yrs | Cross sectional | Median 14.3 (IQR 14.1–14.5) | 55% | 282 | 13.1% (9–17.6%) | eGFR<90ml/min/1.73m$^2$ |
| | | | | | | | 209 | 7.2% (4–11.4%) | Proteinuria of 1+ |
| **Tadesse 2019 [51]** | Ethiopia | Eastern | Perinatally HIV+ < 18 yrs on ART> 6 months | Cohort | Median 12 years (IQR 8–14) | 46.7% | 784 | 0.8% (0.2–1%) | eGFR60-90ml/min/1.73m$^2$ |
| **Mosten 2015 [49]** | Tanzania | Eastern | HIV+ below 18 yrs. | Cross sectional | Mean 10 years (4–18) | 44.8% | 330 | 28.8% (23.9–34%) | Microalbuminuria 20–200 mg/L |
| **Zimba 2015 [55]** | Zambia | Southern | HIV+ 18 months to 16 years on ART | Cross sectional | Mean 9.3 years (SD 3.84) | 50.2% | 209 | 3.8% (1.7–7.4%) | eGFR<60ml/min/1.73m$^2$ |
| | | | | | | | 209 | 8.1% (4.8–12.7%) | Proteinuria of 1+ |
| **Bagoloire 2023 [52]** | Uganda | Eastern | HIV+ 3–17 yrs on ART | Cross sectional | Mean 12.0 (SD 3.21) | 51% | 205 | 10.2 (6.8–15.2) | eGFR<60ml/min/1.73m$^2$ |
| **Areprekumor 2023 [56]** | Nigeria | Western | HIV+ | Cross sectional | 11-18yrs | 50.7% | 150 | 28.1 (17.5–40.8) | Microalbuminuria |
| **Byers 2023 [48]** | Zimbabwe | Southern | HIV+ ART naive | Cohort | 12–17 years | 55% | 266 | 3.8 (1.8–6.8) | eGFR60-90ml/min/1.73m$^2$ |

F: Female, ACR: Albumin creatinine ratio, ART: Antiretroviral therapy, eGFR: Estimated glomerular filtration rate, IQR: Interquartile range, HIV: Human Immunodeficiency Virus, TDF: Tenofovir Disoproxil Fumarate, UNGAL: Urine Neutrophil Gelatinase-Associated Lipocalin. NA: Not applicable MDRD: Modification of Diet in Renal Disease.

infected [6,33,51]. Among the studies that recruited ART-experienced participants, two included only those on TDF-containing regimens [46,53]. Four studies enrolled HIV-negative control groups from the general population [6,33,56,57].

## Diagnosis of CKD

Different methods were used to diagnose CKD. Only one study used the standard definition of CKD with an eGFR cutoff of <60ml/min/1.73m$^2$ on two occasions three or more months apart [55]. The most common biomarker used for CKD diagnosis was serum creatinine measured by the enzymatic method. Estimated GFR (eGFR) was used in 13 studies alone and in combination with markers of kidney damage [6,33,45–48,51–53,55–58]. Five studies used an eGFR cut-off of <60ml/min/1.73m$^2$, [46,47,52,55,57] two used this cutoff alone without any marker of kidney damage [46,57] and six used a cut-off of <90ml/min/1.73m$^2$ [6,33,48,51,53,58]. These studies showed a CKD prevalence between 0.8% and 35.9%. The remaining studies used albumin creatinine ratio (ACR) or microalbuminuria to diagnose CKD [49,50,56]. These studies used ACR >20mg/g [49] or ACR >30 mg/g [50] and a point of care test [56]. Three studies used both ACR and eGFR [33,45,57]. Two studies used proteinuria on dipstick as one of the diagnostic criteria on spot urine samples [54,55]. No study used measured GFR (mGFR) to diagnose CKD. Only four studies repeated the tests to determine chronicity, at week 6 [49], month 3 [55], month 6 [51], and after an average of four years [48]. Equations used to estimate GFR included modified Schwartz in six studies [6,45,46,53,55,57], Modification of Diet in Renal Disease (MDRD) in two studies [47,51], and the full-age spectrum formula in two studies [48,58]. The equation was not mentioned in the remaining three studies.

## Synthesis of results

The pooled CKD prevalence was 12.0% (95% CI 6.0–19.5%), ranging from 0.8% to 53.1%, and was highly heterogeneous (I$^2$ = 97.7%) (Fig 2). Subgroup analysis (Table 2) showed that the lowest pooled CKD prevalence of 2.4% (95% CI 0.6–5.3%) was in the studies that repeated the measurements. The highest pooled CKD prevalence was 27.9% (95% CI 16.7–40.8%) was in the studies that used probability sampling but heterogeneity was high (I$^2$ = 92.6%).

**Assessment of heterogeneity.** Meta-regression analysis results are summarized in Table 3. In univariable meta-regression analysis, the heterogeneity was lower among studies that repeated the eGFR measurements to determine chronicity compared to studies that did not repeat the measurements (β = 0.14 95% CI (0.06–0.24) p value 0.004. The other variables were not statistically significant sources of the observed heterogeneity. In a multivariable meta-regression analysis after adjusting for region, whether or not the measurements were repeated, and the CKD definition used, only whether or not the measurements were repeated accounted for the observed statistical heterogeneity. Studies that repeated measurements for CKD were highly heterogenous compared to studies that did not repeat CKD measurements (adjusted β = 0.11 95% CI (0.15, 0.22) p value 0.03.

**Publication bias.** There was no evidence of publication bias as the funnel plot was symmetrical (Fig 3), and this was confirmed with Egger's test (p = 0.14).

**Risk of bias within and across studies.** Based on the JBI checklist (Table 4), all of the studies had an appropriate sample frame and sampling (n = 16). Only five studies described the study setting and population appropriately. All studies used different criteria to diagnose CKD (Table 1), but the statistical analysis was appropriately performed (n = 16). Since none of the studies scored less than 50%, all were included in the analysis.

**Sensitivity analysis results.** Sensitivity analysis showed that pooled CKD prevalence fell within the 95% confidence interval of the original pooled CKD prevalence when the leave-one-out Jacknife analysis was performed. This suggests that the findings were robust to methodological and analytic approach and that no single study had significant influence on the overall meta-analytic results.

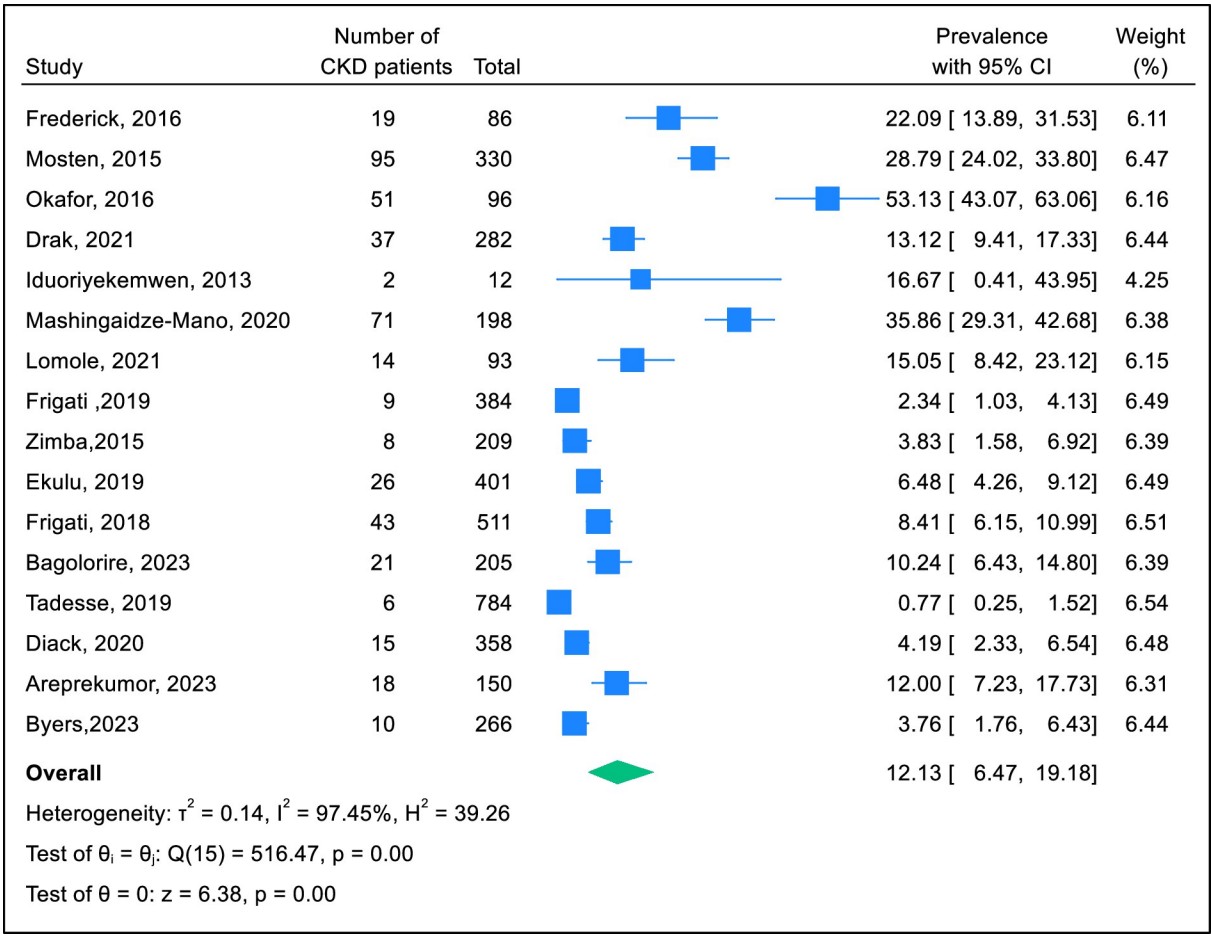

**Fig 2. Forest plot of studies reporting the prevalence of CKD among YPLHIV in SSA.**

## Discussion

To our knowledge, this is the first systematic review and meta-analysis to systematically review CKD prevalence among YPLHIV in sub-Saharan Africa. The pooled prevalence was 12.0%, ranging from 0.8% to 53% with considerable heterogeneity. Findings from subgroup analysis showed that the CKD prevalence differed according to the study sample size, diagnostic definitions, laboratory methods, the region, sampling strategy, and whether or not measurements were repeated to determine chronicity. Only whether or not measurements were repeated accounted for the observed heterogeneity.

There are very few papers that estimate CKD prevalence among YPLHIV in SSA. Systematic reviews and studies on the prevalence of CKD in PLHIV in other settings have also found a variable prevalence of 6 to 48% depending on the region and definitions used [59–62]. In populations at increased risk for CKD such as people living with diabetes, hypertension, sickle cell disease, and PLHIV, a higher CKD prevalence has been reported [12]. In our review, studies included PLHIV and excluded participants with conditions known to cause CKD, such as hypertension, diabetes, sickle cell disease, and hepatitis B and C [63]. This ensured that any observed CKD was probably a result of the various mechanisms by which HIV affects the kidneys, thus reflecting the prevalence of CKD in YPLHIV.

**Table 2. Pooled prevalence of CKD and sources of heterogeneity in the sub-group analysis.**

| Variable | No of studies | Population | Pooled prevalence (%) (95% CI*) | Heterogeneity I² (p value) |
|---|---|---|---|---|
| **Definition of CKD** | | | | |
| eGFR <60ml/min/1.73 m² | 4 | 1173 | 5.9 (3.6–8.7) | 70.6 (0.02) |
| eGFR <90ml/min/1.73m² | 5 | 1962 | 10.4 (1.03–27.5) | 98.9 (<0.001) |
| eGFR & proteinuria | 3 | 390 | 26.9 (5.7–55.4) | 97.5 (<0.001) |
| Albumin Creatinine Ratio | 3 | 747 | 13.1 (6.5, 21.5) | 85.2 (<0.001) |
| **Measurements repeated** | | | | |
| Yes | 3 | 1259 | 2.4 (0.6–5.3) | 83.4 (<0.001) |
| No | 12 | 3013 | 15.4 (8.2–24.4) | 97.2 (<0.001) |
| **Region** | | | | |
| East Africa | 4 | 1405 | 12.8 (2.3–29.7) | 98.1 (<0.001) |
| South Africa | 6 | 1850 | 9.2 (2.6–19.1) | 97.4 (<0.001) |
| West Africa | 4 | 616 | 18.7 (2.5–43.7) | 97.0 (<0.001) |
| Central Africa | 1 | 401 | 6.5 (4.2–9.1) | - |
| **Participants** | | | | |
| <100 | 3 | 194 | 31.6 (11.6–55.5) | 88.7 (<0.001) |
| 100–300 | 6 | 1310 | 11.6 (4.6–21.2) | 95.6 (<0.001) |
| >300 | 6 | 2768 | 6.7 (1.6–14.7) | 97.9 (<0.001) |
| **By publication year** | | | | |
| Before 2019 | 6 | 1244 | 19.9 (7.4–36.4) | 97.1 (<0.001) |
| After 2019 | 9 | 3028 | 8.0 (3.2–14.7) | 97.0 (<0.001) |
| **ART status at enrolment** | | | | |
| ART naïve | 3 | 644 | 19.6 (0.8–53.4) | 98.7 (<0.001) |
| On ART | 10 | 3213 | 7.8 (3.1–14.1) | 96.7 (<0.001) |
| Both naïve and on ART | 2 | 416 | 26.7 (3.8–32.8) | 33.7 (0.22) |
| **Sampling strategy** | | | | |
| Probability | 6 | 872 | 27.9 (16.7–40.8) | 92.6 (<0.001) |
| Nonprobability | 4 | 1233 | 5.3 (1.9–10.2) | 90.7 (<0.001) |
| Not mentioned | 5 | 2167 | 5.2 (2.1–9.6) | 93.2 (<0.001) |

*CI confidence interval.

KDIGO defines CKD as GFR<60ml/min/1.73m² with or without markers of kidney damage for three months or more [12] and this definition has been shown to improve diagnostic precision with implications for management [64]. However, only four studies fulfilled the chronicity criteria; three studies repeated the tests after more than 3 months [48,51,55], and one study repeated the tests after only one month [49]. Three studies used a cut-off of <90 ml/min/1.73m² which does not fit the KDIGO definition and tended to overestimate the CKD prevalence [6,53,54]. Other studies used ranges such as 60–90 or 30–60 ml/min/1.73m² potentially underestimating the CKD prevalence [46,51]. Albuminuria is known as an early marker of endothelial and kidney damage and might precede a decrease in GFR in people with CKD [33]. Varying cut-offs of albuminuria and proteinuria were also used in the studies. Using these in isolation has limitations as several conditions (acute kidney injury, febrile illness, and intense exercise) lead to transient proteinuria hence an overestimate of the CKD prevalence [65]. This limitation has also been mentioned in previous systematic reviews of CKD prevalence in Africa among the general population and other high-risk groups [61]. Two of the studies with CKD prevalence exceeding 20% used albuminuria for CKD diagnosis [49,50].

**Table 3. Factors associated with CKD prevalence using univariable and multivariable meta-regression analysis findings.**

| Variable | Univariable meta-regression analysis | | Multivariable meta-regression analysis | |
| --- | --- | --- | --- | --- |
| | Coefficient (95% CI*) | P value | Coefficient (95% CI*) | P value |
| **Definition of CKD** | | | | |
| eGFR & proteinuria | 0 | | 0 | |
| eGFR <60ml/min/1.73 m$^2$ | 0.02 (-0.13,018) | 0.73 | 0.03 (-1.10, 0.17) | 0.61 |
| eGFR <90ml/min/1.73m$^2$ | -0.04 (-0.19, 0.09) | 0.47 | -0.07 (-0.19, 0.05) | 0.26 |
| Albumin Creatinine Ratio | 0.06 (-0.10, 0.22) | 0.44 | 0.03 (-0.09, 0.16) | 0.65 |
| **Measurements repeated** | | | | |
| No | 0 | | 0 | |
| Yes | 0.14 (0.06–0.24) | 0.004 | 0.11 (0.15, 0.22) | 0.03 |
| **SSA Region** | | | | |
| West Africa | 0 | | 0 | |
| Central Africa | 0.04 (-0.16, 0.25) | 0.62 | 0.03 (-0.10, 0.16) | 0.65 |
| East Africa | -0.02 (-0.16, 0.13) | 0.81 | 0.08 (-0.03, 0.21) | 0.15 |
| South Africa | 0.16 (-0.12, 0.155) | 0.79 | 0.07 (-0.03, 0.211) | 0.16 |
| **Sample size** | | | | |
| >300 | 0 | | - | - |
| 100–300 | 0.05 (-0.05, 0.14) | 0.29 | - | - |
| <100 | -0.02 (-0.23, 0.19) | 0.85 | - | - |
| **Publication year** | | | | |
| After 2019 | 0 | | - | - |
| Before 2019 | 0.03 (-0.8, 0.13) | 0.61 | - | - |
| **ART status at enrolment** | | | | |
| On ART | 0 | | - | - |
| Both naïve and on ART | 0.06 (-0.10, 0.22) | 0.48 | - | - |
| ART naïve | 0.01 (-0.12, 0.13) | 0.90 | - | - |
| **Sampling strategy** | | | | |
| Nonprobability | | | | |
| Probability | 0.04 (-0.14, 0.24) | 0.62 | - | - |
| Not mentioned | -0.01 (-0.12,0.10) | 0.84 | - | - |

CI confidence interval, CKD Chronic kidney disease, eGFR estimated glomerular filtration rate, SSA Sub Saharan Africa, ART Anti-retroviral therapy.

The most common biomarker used to estimate GFR was serum creatinine despite its limitations [66] such as being affected by muscle mass, general nutritional status, exercise, and diet [31]. Using creatinine to estimate GFR has been shown to overestimate GFR, potentially underestimating CKD in a large multicentre study conducted in Uganda, Malawi, and South Africa [67]. Most studies used the Jaffe method to measure serum creatinine which has also been shown to overestimate creatinine clearance compared to the enzymatic method [61] leading to low CKD prevalence as diagnosed by eGFR. Cystatin C has been reported as a better biomarker in Africa but is rarely used because it is expensive [67] and none of the studies included used Cystatin C. One of the studies used UNGAL as the biomarker, this has shown promise for early diagnosis of CKD as it is elevated before decline in eGFR and development of albuminuria [68]. Equations that use multiple markers are also better at estimating GFR than those that use a single marker [69,70] but all included studies used a single marker. Further, none of the included studies measured GFR, yet it could have been useful given the limitations of serum creatinine [71,72].

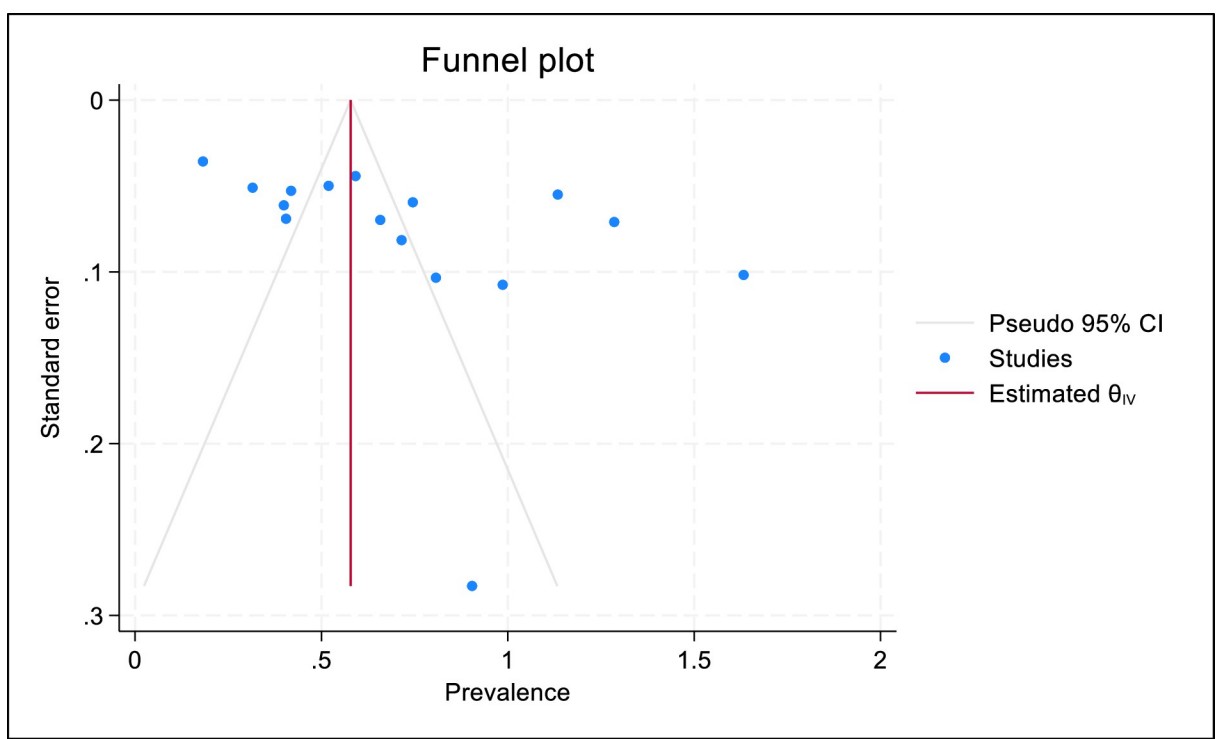

**Fig 3. Funnel plot of included studies.**

**Table 4. Results of the bias assessment using the JBI checklist.**

| Author, year | Qn1* | Qn2 | Qn3 | Qn4 | Qn5 | Qn6 | Qn7 | Qn8 | Qn9 | Score | Overall appraisal |
|---|---|---|---|---|---|---|---|---|---|---|---|
| **Diack, 2020** [46] | Yes | No | No | No | Yes | Yes | Yes | Yes | Yes | 0.7 | Include |
| **Okafor, 2016** [47] | Yes | No | No | Yes | Yes | Yes | Yes | Yes | Yes | 0.8 | Include |
| **Ekulu, 2019** [57] | Yes | No | No | No | Yes | Yes | Yes | No | Yes | 0.6 | Include |
| **Frigati, 2019** [6] | Yes | No | No | No | Yes | Yes | Yes | Yes | Yes | 0.7 | Include |
| **Frederick, 2016** [50] | Yes | Unk | Yes | No | Yes | No | Yes | Yes | Yes | 0.7 | Include |
| **Frigati, 2018** [33] | Yes | Unk | Unk | No | Yes | Yes | Yes | Yes | Yes | 0.7 | Include |
| **Iduoriyekemwen, 2013** [45] | Yes | Unk | Unk | No | Yes | Yes | Yes | Yes | Yes | 0.7 | Include |
| **Mashingaidze-Mano, 2020** [53] | Yes | No | No | No | Yes | Yes | Yes | Yes | Yes | 0.7 | Include |
| **Drak 2021** [54] | Yes | Unk | Unk | Yes | Yes | Yes | Yes | Yes | Yes | 0.8 | Include |
| **Tadesse, 2019** [51] | Yes | Unk | Unk | Yes | Yes | Yes | Yes | Yes | Yes | 0.8 | Include |
| **Mosten, 2015** [49] | Yes | Yes | Unk | No | Yes | Yes | Yes | Yes | Yes | 0.8 | Include |
| **Zimba, 2015** [55] | Yes | Yes | Yes | Yes | Yes | Yes | Yes | Yes | Yes | 1 | Include |
| **Bagoloire, 2023** [52] | Yes | Unk | Yes | Yes | Yes | Yes | No | Yes | Unk | 0.7 | Include |
| **Areprekumor, 2023** [56] | Yes | No | Yes | Yes | Yes | No | No | Yes | Unk | 0.6 | Include |
| **Byers, 2023** [48] | Yes | Unk | Unk | Yes | Yes | Yes | Yes | Yes | Unk | 0.7 | Include |

Unk Unknown *1. Was the sample frame appropriate to address the target population?2.Were study participants sampled in an appropriate way?3.Was the sample size adequate?4.Were the study subjects and the setting described in detail?5.Was the data analysis conducted with sufficient coverage of the identified sample?6.Were valid methods used for the identification of the condition?7.Was the condition measured in a standard, reliable way for all participants?8.Was there appropriate statistical analysis?9.Was the response rate adequate, and if not, was the low response rate managed appropriately?

Estimating equations such as the Modification of Diet in Renal Disease (MDRD), Chronic Kidney Disease Epidemiology (CKD-EPI), and Cockcroft-Gault (CG) equations are frequently used to estimate GFR [16]. These equations have been validated in many diverse populations, but the estimated GFR falls short of the measured GFR [66,67,71], and accuracy is worse when the coefficients for race are added into the equation [71]. A large multicentre study in Uganda, Malawi, and South Africa found that none of the estimating equations they assessed, including the CKD-EPI 2021, gave a GFR that was within 30% of the measured GFR for more than 75% of the samples [67]. The review argued that there is an urgent need to find an affordable and sensitive biomarker to overcome the current diagnostic inaccuracies [67,72] and to develop an equation for African populations [73].

Variability in CKD prevalence has also been reported in other populations across the world. Two systematic reviews that determined the CKD prevalence in the general African population found a prevalence similar to ours of 13.9% (95% CI 12.2–15.7) [18] but 2% to 41% in the general population and 1% to 46% among PLHIV [61]. In adults living with HIV, a systematic review and meta-analysis across the globe found a pooled CKD prevalence of 6.4% (95% CI 5.2–7.7%), with high heterogeneity ($I^2$ of 99.2% p<0.001) that was explained by the World Health Organisation region [59]. Younger children living with HIV aged 1 to 18 years have a variable and high CKD prevalence as well, ranging from 6.7% to 34.6% in African studies. A study in Zimbabwe among 220 ART-naïve children aged 2–12 years found a CKD prevalence of 34.6% (95% CI 27.8–40.8%) using eGFR 30-90ml/l/1.73m$^2$ [74]. Children and adolescents aged 1 month to 18 years who were mostly ART naïve (86%) have been found to have a microalbuminuria prevalence of 12% (95% CI 4.5–24.3%) in Nigeria [75]. Another Nigerian study involving 60 children who estimated GFR using cystatin C found a CKD prevalence of 13.3% (95% CI 5.9–24.6%) with an eGFR cut-off of less than 60ml/min/1.73 m$^2$ [76].

## Strengths and limitations

Strengths of our study include being the first to systematically report CKD prevalence among YPLHIV in SSA, no evidence of publication bias, findings being robust to the analytic approach and methodology, and the search strategy being sensitive and comprehensive. The major limitation is that all but one of the included studies did not follow the standard definition of CKD, and the diagnostic criteria varied widely. Other limitations were a lack of age-disaggregation in the reporting of the results in some studies that led to the exclusion of 34 papers.

## Conclusions and recommendations

The CKD prevalence among YPLHIV across SSA countries is moderately high and highly heterogeneous. The use of standardized definitions and diagnostic methods is urgently needed to improve the CKD prevalence estimates and to improve the precision of the pooled estimate. Better reporting is also needed to detail the methods used and to disaggregate CKD prevalence by age to isolate the most affected age groups. The moderately high CKD prevalence implies that HIV control programs need to routinely screen YPLHIV for CKD to ensure early diagnosis and management hence improving survival and quality of life in this young population.

## Supporting information

**S1 Appendix. PRISMA checklist.**
(DOCX)

**S2 Appendix. Study protocol.**
(DOCX)

**S3 Appendix. Search strategy.**
(DOCX)

**S4 Appendix. Details of the included studies.**
(DOCX)

**S5 Appendix. Search results.**
(XLSX)

## Acknowledgments

The authors would like to acknowledge James Prior for the support to EMN and the Librarian at LSHTM, Kate Perris who helped with the development and review of the search terms.

## Author Contributions

**Conceptualization:** Esther M. Nasuuna, Chido Dziva Chikwari, Robert Kalyesubula, Laurie A. Tomlinson, Helen A. Weiss.

**Data curation:** Esther M. Nasuuna, Nicholus Nanyeenya, Davis Kibirige, Jonathan Izudi.

**Formal analysis:** Esther M. Nasuuna, Nicholus Nanyeenya, Davis Kibirige, Jonathan Izudi.

**Funding acquisition:** Barbara Castelnuovo.

**Methodology:** Esther M. Nasuuna, Nicholus Nanyeenya, Jonathan Izudi, Laurie A. Tomlinson, Helen A. Weiss.

**Project administration:** Barbara Castelnuovo.

**Resources:** Laurie A. Tomlinson.

**Supervision:** Chido Dziva Chikwari, Robert Kalyesubula, Barbara Castelnuovo, Laurie A. Tomlinson, Helen A. Weiss.

**Validation:** Helen A. Weiss.

**Writing – original draft:** Esther M. Nasuuna.

**Writing – review & editing:** Nicholus Nanyeenya, Davis Kibirige, Jonathan Izudi, Chido Dziva Chikwari, Robert Kalyesubula, Barbara Castelnuovo, Laurie A. Tomlinson, Helen A. Weiss.

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
