## [Decision Letter · Decision Letter 0]

27 May 2024

PONE-D-24-11592Prevalence of chronic kidney disease among young people living with HIV in Sub Saharan Africa: A systematic review and meta-analysis.PLOS ONE

Dear Dr. Nasuuna,

Thank you for submitting your manuscript to PLOS ONE. After careful consideration, we feel that it has merit but does not fully meet PLOS ONE’s publication criteria as it currently stands. Therefore, we invite you to submit a revised version of the manuscript that addresses the points raised during the review process.

We look forward to receiving your revised manuscript.

Kind regards,

Udeme Ekpenyong Ekrikpo, MBBS PhD

Academic Editor

PLOS ONE

“The authors would like to acknowledge James Prior for the support to EMN and the Librarian at LSHTM, Kate Perris who helped with the development and review of the search terms. Support for research was provided by Fogarty International Centre, National Institutes of Health (grant #2D43TW009771-06) HIV and co-infections in Uganda. HAW is funded by the UK Medical Research Council (MRC) and the UK Department for International Development (DFID) under the MRC/DFID Concordat agreement (Grant Ref: MR/R010161/1). EN, Doctoral Research Fellow, NIHR131273 is funded by the NIHR for this research project. The views expressed in this publication are those of the authors and not necessarily those of the NIHR, NHS or the UK Department of Health

and Social Care.”

Additional Editor Comments:

1. The authors should pay attention to the poor formatting of all in-text citations. Please ensure proport formatting using the PLOS ONE citation style.

2. The primary weakness of this manuscript is the authors' failure to use a standard definition for CKD as prescribed by KDIGO. The included studies utilized varying definitions of kidney dysfunction from dipstick proteinuria of at least +1, albumin-creatinine ratio and varying cut-off of eGFR. Also, it appears the cross-sectional studies reported single measures of eGFR and/or ACR. This makes it possible that individuals with AKI were included in this analysis and wrongly classified as CKD. The authors can circumvent this by including a sub-group meta-analysis of studies that employed the strict KDIGO definition of CKD.

3. Please report a sub-group analysis comparing the kidney dysfunction prevalence in the ARV-naive compared to the ARV-exposed population.

4. 13/16 (81.3%) and not the reported 86.7% of studies were cross-sectional. 

Reviewers' comments:

Reviewer's Responses to Questions

**Comments to the Author**

1. Is the manuscript technically sound, and do the data support the conclusions?

Reviewer #1: Partly

Reviewer #2: Partly

Reviewer #3: Partly

2. Has the statistical analysis been performed appropriately and rigorously? 

Reviewer #1: Yes

Reviewer #2: Yes

Reviewer #3: Yes

3. Have the authors made all data underlying the findings in their manuscript fully available?

Reviewer #1: Yes

Reviewer #2: Yes

Reviewer #3: Yes

4. Is the manuscript presented in an intelligible fashion and written in standard English?

Reviewer #1: Yes

Reviewer #2: Yes

Reviewer #3: Yes

5. Review Comments to the Author

Reviewer #1: The definition of CKD used lacked standardization, hence the likelihood for inclusion of non-CKD and AKI cases. Also why was the study that used UNGAL included, as it doesn't differentiate CKD from AKI.

Reviewer #2: All the included articles in this study did not use the standard definition of CKD. This makes the validity of the study doubtful based on the primary objective of the study. I also acknowledged that this was stated as a limitation. The authors may consider to include only articles that used the standard definition of CKD or may change the description of their study population from CKD to a term that will reflect the population included in this study which is not strictly CKD population

Reviewer #3: The topic is challenging but should using the standard CKD definition eGFR < 60 ml/min/1.73m with marker of kidney damage for 3 months or more to choose the eligible studies.

line 35 wrong YPHLIV abbreviation, line 43 wrong %, line 80 phosphaturia not a marker and need to rewrite the whole sentence in proper way.

line 83 should include Africa %. line 90 need to rewrite the whole sentence in proper way.

line 98 mention the extend of higher risk (fold).

box 1 use the original reference of KDIGO.

line 190 IOM Abb. came late.

line 195 "fulfilled the inclusion criteria" wrongly used as the 802 not fulfilled the criteria.

line 253 using or not proper the same Abb.

Table 2 title sample size (4th sector) is wrong.

line 299 the funnel plot shows Asymmetry and Egger test is not sig. should use the Pegg test.

line 352 albumin not used to diagnose CKD should be Albuminuria.

In the whole manuscript there are Invalid:

- format of the references

- full stop after the Headings

- using comma in 95% CI range

6. PLOS authors have the option to publish the peer review history of their article (what does this mean?). If published, this will include your full peer review and any attached files.

Reviewer #1: No

Reviewer #2: No

Reviewer #3: No

---

## [Author Response · Author response to Decision Letter 0]

17 Jun 2024

Response to editor and reviewers. 

Author comment

I thank the editor and the reviewers for the considered comments that have greatly improved this paper. The editor and the reviewers highlight an important point about the papers that were included in the systematic review not following the standard definition of CKD as recommended by KDIGO. 

Although, we found only one paper that followed the strict KDIGO criteria of GFR <60ml/min/1.73m2, we still found it prudent to report the broader results of the comprehensive systematic review to show that there is very limited research and what has been done does not follow the gold-standard definitions of CKD. Thus there is a substantial evidence gap regarding kidney health among young people living with HIV in Africa. Given the prevalence of HIV on the continent and the young median age of the population of many countries in Africa this is a critical area to understand the potential future burden of CKD.

Response: The manuscript has been formatted according to the templates above. 

Response: The colleague that edited my manuscript is Helen Weiss.

“The authors would like to acknowledge James Prior for the support to EMN and the Librarian at LSHTM, Kate Perris who helped with the development and review of the search terms. Support for research was provided by Fogarty International Centre, National Institutes of Health (grant #2D43TW009771-06) HIV and co-infections in Uganda. HAW is funded by the UK Medical Research Council (MRC) and the UK Department for International Development (DFID) under the MRC/DFID Concordat agreement (Grant Ref: MR/R010161/1). EN, Doctoral Research Fellow, NIHR131273 is funded by the NIHR for this research project. The views expressed in this publication are those of the authors and not necessarily those of the NIHR, NHS or the UK Department of Health and Social Care.”

Response: All funding-related information has been removed from the acknowledgements section. The cover letter has a section with funding statement for the revision. 

Additional Editor Comments:

1. The authors should pay attention to the poor formatting of all in-text citations. Please ensure proport formatting using the PLOS ONE citation style.

Response: The references have been formatted as per the PloS One citation style. 

2. The primary weakness of this manuscript is the authors' failure to use a standard definition for CKD as prescribed by KDIGO. The included studies utilized varying definitions of kidney dysfunction from dipstick proteinuria of at least +1, albumin-creatinine ratio and varying cut-off of eGFR. Also, it appears the cross-sectional studies reported single measures of eGFR and/or ACR. This makes it possible that individuals with AKI were included in this analysis and wrongly classified as CKD. The authors can circumvent this by including a sub-group meta-analysis of studies that employed the strict KDIGO definition of CKD.

Response: Of the four studies that had repeated measurements of GFR, only one (Zimba 2016) followed the strict KDIGO definition of CKD of GFR<60ml/min/1.73m2 repeated after 3 months. Tadesse 2019 used a definition of <90ml/min/1.73m2 and Byers 2023 used a definition of <90ml/min/1.73m2. Mosten 2015 diagnosed CKD using albuminuria and repeated the urinalysis after only one month. It is, therefore, unfortunately not possible to do a meaningful subgroup analysis. 

3. Please report a sub-group analysis comparing the kidney dysfunction prevalence in the ARV-naive compared to the ARV-exposed population.

Response: This is provided in Table 2 at the top of page 12. 

4. 13/16 (81.3%) and not the reported 86.7% of studies were cross-sectional. 

Response: We thank the reviewer for this correction. This has been changed on line 43

Comments to the Author

5. Review Comments to the Author

Reviewer #1: The definition of CKD used lacked standardization, hence the likelihood for inclusion of non-CKD and AKI cases. Also why was the study that used UNGAL included, as it doesn't differentiate CKD from AKI.

Response: Thank you for this observation. As the reviewer rightly states, the studies used different definitions to diagnose CKD in their populations. We have stated that only one following the KDIGO recommended definition in the results (line 67-68). In the Discussion (lines 354-356), we explain that UNGAL is one of the newer biomarkers that might be able to help diagnose kidney dysfunction earlier than the traditionally used ones. Line 351-352. Although this study did not use creatinine based GFR, we chose to show what has been done to show kidney function among YPLHIV in SSA and it was better to be more inclusive than exclusive. However, in this study (as in the other 12 studies), they did not repeat the measurements to determine chronicity. We include this as a major limitation in this study. 

Reviewer #2: All the included articles in this study did not use the standard definition of CKD. This makes the validity of the study doubtful based on the primary objective of the study. I also acknowledged that this was stated as a limitation. The authors may consider to include only articles that used the standard definition of CKD or may change the description of their study population from CKD to a term that will reflect the population included in this study which is not strictly CKD population

Response: We thank you for this comment, this concern has been addressed in the author comment at the top of page 1 in this response to reviewer comment. 

Reviewer #3: The topic is challenging but should using the standard CKD definition eGFR < 60 ml/min/1.73m with marker of kidney damage for 3 months or more to choose the eligible studies. 

Response: Thank you for this suggestion. The reasons for not following the standard definition of CKD have been discussed in the author comments above. 

line 35 wrong YPHLIV abbreviation, 

Response: Thank you for picking this up. This has been corrected. Line 35

line 43 wrong %, 

Response: Thank you for bringing this to our attention. This has been corrected. Line 43

line 80 phosphaturia not a marker and need to rewrite the whole sentence in proper way.

Response: Thank you for this comment. Phosphaturia has been removed as a marker of kidney damage. Line 75

line 83 should include Africa %. 

Response: Thank you for the suggestion. The prevalence of CKD in Sub-Saharan Africa has been added. Lines 78-79. 

line 90 need to rewrite the whole sentence in proper way.

Response: Thank you for picking this up. This has been rewritten. Lines 85 to 86

line 98 mention the extend of higher risk (fold).

Response: Thank you for the suggestion. A fourfold higher risk of kidney disease has been added. Lines 92. 

box 1 use the original reference of KDIGO.

Response: Thank you. The original reference of KDIGO has been added to the box (Reference 1, Box 1, fourth line). 

line 190 IOM Abb. came late.

Response: Thank you for bringing this to our attention. We have removed this abbreviation (line 178)

line 195 "fulfilled the inclusion criteria" wrongly used as the 802 not fulfilled the criteria.

Response: Thank you. We have reworded this to indicate articles that fulfilled the search criteria (lines 184 to 185). 

line 253 using or not proper the same Abb.

Response: Thank you. We have corrected this (line 222). 

Table 2 title sample size (4th sector) is wrong.

Response: Thank you for noticing this. This has been changed to population in Table 2. 

line 299 the funnel plot shows Asymmetry and Egger test is not sig. should use the Pegg test.

Response: Thank you for the suggestion. The funnel plot may look asymmetrical, but it is not very conclusive to determine if there is publication bias simply by looking at the funnel plot especially if the review has few studies as in this case. The Begg test in this analysis showed a p value of 0.03 but we think publication bias is unlikely given the substantial heterogeneity of the studies in this review, and prefer to retain the Egger test as it has been shown to be more accurate (1).”

line 352 albumin not used to diagnose CKD should be Albuminuria.

Response: Thank you for bringing this to our attention. We have changed this to albuminuria (line 318).

In the whole manuscript there are Invalid: Thank you for pointing these out. They have been addressed as shown below. 

- format of the references: These have been reformatted throughout the manuscript. 

- full stop after the Headings: All full stops have been removed from the headings. 

- using comma in 95% CI range: All commas have been removed from the 95% CI range

References

1. Shi X, Nie C, Shi S, Wang T, Yang H, Zhou Y, et al. Effect comparison between Egger’s test and Begg’s test in publication bias diagnosis in meta-analyses: evidence from a pilot survey. Int J Res Stud Biosci. 2017;5(5):14-20.

---

## [Editor Report · Decision Letter 1]

29 Jul 2024

PONE-D-24-11592R1Prevalence of chronic kidney disease among young people living with HIV in Sub Saharan Africa: A systematic review and meta-analysis.PLOS ONE

Dear Dr. Nasuuna,

Thank you for submitting your manuscript to PLOS ONE. After careful consideration, we feel that it has merit but does not fully meet PLOS ONE’s publication criteria as it currently stands. Therefore, we invite you to submit a revised version of the manuscript that addresses the points raised during the review process.

The author's response to the reviewer's comments is noted. The limitations of this manuscript are also well noted.

Please do the following and re-submit the manuscript.

1. Please remove the urinary NGAL study from the list of articles included, as suggested by one of the reviewers. Using an NGAL study simultaneously with creatinine-based GFR does not appear appropriate. Repeat the analysis without the NGAL study. Of course, the NGAL study can be referenced in the discussion section while highlighting the need for early CKD diagnosis.

2. Kindly convert Appendix 4 from EXCEL to MS Word format and resubmit.

We look forward to receiving your revised manuscript.

Kind regards,

Udeme Ekpenyong Ekrikpo, MBBS PhD

Academic Editor

PLOS ONE

---

## [Author Response · Author response to Decision Letter 1]

31 Jul 2024

Please do the following and re-submit the manuscript.

1. Please remove the urinary NGAL study from the list of articles included, as suggested by one of the reviewers. Using an NGAL study simultaneously with creatinine-based GFR does not appear appropriate. Repeat the analysis without the NGAL study. Of course, the NGAL study can be referenced in the discussion section while highlighting the need for early CKD diagnosis.

Response: the NGAL paper has been removed from the analysis. The systematic review now includes 15 and not 16 papers. This has been reflected in all tables, figures and in the text. See attached manuscript. 

2. Kindly convert Appendix 4 from EXCEL to MS Word format and resubmit

Response: Appendix 4 has been converted to word and is attached as a word document.

---

## [Editor Report · Decision Letter 2]

7 Aug 2024

Prevalence of chronic kidney disease among young people living with HIV in Sub Saharan Africa: A systematic review and meta-analysis.

PONE-D-24-11592R2

Dear Dr. Nasuuna

We’re pleased to inform you that your manuscript has been judged scientifically suitable for publication and will be formally accepted for publication once it meets all outstanding technical requirements.

Kind regards,

Udeme Ekpenyong Ekrikpo, MBBS PhD

Academic Editor

PLOS ONE
---

## [Editor Report · Acceptance letter]

16 Aug 2024

PONE-D-24-11592R2 

PLOS ONE

Dear Dr. Nasuuna, 

I'm pleased to inform you that your manuscript has been deemed suitable for publication in PLOS ONE. Congratulations! Your manuscript is now being handed over to our production team.

Kind regards, 

on behalf of

Associate Professor Udeme Ekpenyong Ekrikpo 

Academic Editor

PLOS ONE